# Interdependent Regulation of Polycystin Expression Influences Starvation-Induced Autophagy and Cell Death

**DOI:** 10.3390/ijms222413511

**Published:** 2021-12-16

**Authors:** Jean-Paul Decuypere, Dorien Van Giel, Peter Janssens, Ke Dong, Stefan Somlo, Yiqiang Cai, Djalila Mekahli, Rudi Vennekens

**Affiliations:** 1Laboratory of Pediatrics, PKD Research Group, Department of Development and Regeneration, KU Leuven, 3000 Leuven, Belgium; dorien.vangiel@kuleuven.be (D.V.G.); peter.janssens@uzbrussel.be (P.J.); djalila.mekahli@uzleuven.be (D.M.); 2Laboratory of Ion Channel Research, Biomedical Sciences Group, Department of Cellular and Molecular Medicine, KU Leuven, 3000 Leuven, Belgium; rudi.vennekens@kuleuven.be; 3Department of Nephrology, University Hospitals Brussels, 1090 Brussels, Belgium; 4Department of Internal Medicine, Yale School of Medicine, New Haven, CT 06520, USA; ke.dong@yale.edu (K.D.); stefan.somlo@yale.edu (S.S.); yiqaing.cai@yale.edu (Y.C.); 5Department of Genetics, Yale School of Medicine, New Haven, CT 06520, USA; 6Department of Pediatric Nephrology, University Hospital of Leuven, 3000 Leuven, Belgium; 7VIB-KU Leuven Center for Brain & Disease Research, 3000 Leuven, Belgium

**Keywords:** ADPKD, autosomal dominant polycystic kidney disease, autophagy, cell death, nutrient stress, polycystins

## Abstract

Autosomal dominant polycystic kidney disease (ADPKD) is mainly caused by deficiency of polycystin-1 (PC1) or polycystin-2 (PC2). Altered autophagy has recently been implicated in ADPKD progression, but its exact regulation by PC1 and PC2 remains unclear. We therefore investigated cell death and survival during nutritional stress in mouse inner medullary collecting duct cells (mIMCDs), either wild-type (WT) or lacking PC1 (PC1KO) or PC2 (PC2KO), and human urine-derived proximal tubular epithelial cells (PTEC) from early-stage ADPKD patients with PC1 mutations versus healthy individuals. Basal autophagy was enhanced in PC1-deficient cells. Similarly, following starvation, autophagy was enhanced and cell death reduced when PC1 was reduced. Autophagy inhibition reduced cell death resistance in PC1KO mIMCDs to the WT level, implying that PC1 promotes autophagic cell survival. Although PC2 expression was increased in PC1KO mIMCDs, PC2 knockdown did not result in reduced autophagy. PC2KO mIMCDs displayed lower basal autophagy, but more autophagy and less cell death following chronic starvation. This could be reversed by overexpression of PC1 in PC2KO. Together, these findings indicate that PC1 levels are partially coupled to PC2 expression, and determine the transition from renal cell survival to death, leading to enhanced survival of ADPKD cells during nutritional stress.

## 1. Introduction

Autosomal dominant polycystic kidney disease (ADPKD) is the most common cystic kidney disease with a prevalence of ca. 1 in 1000–4000 [1]. Although ADPKD is pauci-symptomatic until the 3rd–4th decade of life, (micro)cysts and symptoms are known to develop in children and even in utero [2]. It represents the fourth most common cause of kidney failure [3] and dialysis and transplantation remain the ultimate therapy. The only approved drug is Tolvaptan, a selective vasopressin receptor 2 antagonist [4], but its unfavorable side effects, including aquaresis leading to polyuria, dictate the need for alternative treatment options [5].

Renal cysts mostly develop due to mutations in *PKD1* (ca. 78%) or *PKD2* (ca. 15%) [6], encoding for polycystin 1 (PC1) and 2 (PC2), respectively. Mutations in *PKD1* generally lead to a faster disease progression with kidney failure at a younger age compared to *PKD2* [7]. PC1 and PC2 co-interact and function as a signaling complex with PC2 being a Ca^2+^-release channel [8,9]. However, the exact regulation and cellular consequences of their activation remain elusive [10]. Loss of function of PC1 and/or PC2 in animal cell lines leads to reduced intracellular Ca^2+^ signaling, increased mTOR activity, increased cAMP levels and increased proliferation [11].

Recently, autophagy has emerged as a possible contributor in ADPKD progression, although the exact nature of its dysregulation in ADPKD remains unresolved [12]. Autophagy was reported to be upregulated, reduced or impaired in various animal models of ADPKD [13,14,15]. PC2 seems to be indispensable for stress-induced autophagy [16,17,18,19], but these studies do not involve renal cells. Moreover, none of these studies considered PC1 and PC2 at the same time. This is nonetheless important as PC1 and PC2 function together [9,20,21]. Therefore, we hypothesized that polycystin-deficient renal cells have altered cell survival and death during stress, and that this is caused by the absence of an interdependent regulation of PC1 and PC2 expression.

## 2. Results

### 2.1. PC1KO mIMCDs Are Partially Resistant against Starvation-Induced Cell Death

To study the role of PC1 in starvation-induced cell death and survival, wild-type (WT) and PC1 knockout (PC1KO) mouse inner medullary collecting duct cells (mIMCDs) were subjected to nutrient starvation, a potent inducer of autophagy, by incubation in Hanks’ balanced salt solution (HBSS). A genomic deletion in exon 2–3 of *PKD1* in PC1KO cells was confirmed (Appendix A). Because mIMCDs are triploid, sequencing resulted in an overlap of the signals of all three *PKD1* copies (Appendix A), which made it impossible to detect the specific mutations in each allele generated by CRISPR-Cas9. However, gel electrophoresis showed a deletion in all copies (Appendix A), protein levels remained undetected (Figure 1A) and mRNA expression was strongly reduced (Figure 1B). Some residual mRNA expression was still observed, most likely because the qPCR primers used target a region mostly upstream (exon 1–2) of the regions targeted by CRISPR-Cas9 (exon 2–3). WT and PC1KO mIMCDs showed similar morphology, proliferation rate and ATP production (Appendix A). Microscopic observation revealed more cell detachment in IMCD WT compared to PC1KO following 80 h of nutrient starvation (Figure 1C). This was supported by a stronger decline in confluence (Figure 1D), as well as more Cytotox-Green-positive signals (Figure 1E) in WT mIMCDs compared to PC1KO. The elevation of cell death in WT mIMCDs compared to PC1KO was confirmed with Trypan Blue (Figure 1F), and was associated with the reduced levels of cleaved Caspase-3 in PC1KO mIMCDs (Figure 1G). Transient overexpression of human PC1 (hPC1) in WT further reduced the number of live cells (Figure 1H). Restoration of PC1 levels by overexpression of hPC1 in PC1KO during starvation also reduced the number of live cells similar to WT cells (Figure 1H), suggesting that the enhanced resistance in PC1KO cells during starvation is dependent on PC1 absence. This could have implications for the recovery following starvation. Indeed, replacing HBSS with normal medium (DMEM) led to an increase in confluence due to enhanced proliferation, which was strongest in PC1KO mIMCDs (Figure 1I). We then evaluated whether the enhanced proliferation was dependent on mTOR, since enhanced mTOR activity is suggested to support cyst formation. Interestingly, the phosphorylation of ribosomal protein S6 did not suggest changes in basal mTOR activity (Figure 1J); while following 48 h of HBSS and 24 h of recovery in DMEM, phosphorylated S6 was significantly higher in PC1KO compared to WT mIMCDs (Figure 1K). Although S6 is indirectly phosphorylated by mTOR (via p70S6K), these results suggest that during recovery, mTOR reactivation is potentially stronger in PC1KO compared to WT mIMCDs.

### 2.2. Increased Basal Autophagy in PC1KO mIMCDs

Since PC1KO mIMCDs are more resistant to starvation-induced death compared to WT cells, we investigated the autophagy survival pathway. We observed more LC3-II in PC1KO compared to WT mIMCDs both in the presence and absence of the lysosomal blocker Baf A1 (Figure 2A), signifying enhanced autophagy. Of note, the visualization of LC3-I expression in mIMCDs required stronger exposure as compared to LC3-II, a characteristic similar to other cell types such as HeLa cells [22]. Furthermore, the autophagy degradation substrate Sqstm1 (also known as p62) was decreased in PC1KO compared to WT mIMCDs, suggesting enhanced autophagic degradation (Figure 2A,B). mRNA expression of LC3 was also significantly increased in PC1KO cells (Figure 2C). In addition, more GFP-LC3 punctae were observed in transiently transfected PC1KO mIMCDs (Figure 2D). Enhanced basal autophagy in PC1KO mIMCDs was reduced by overexpression of hPC1 in PC1KO mIMCDs (Figure 2E), suggesting that basal autophagy is directly dependent on PC1 protein expression. The stronger PC1 band in the overexpressing mIMCDs is likely caused by the enhanced reactivity for human compared to mouse PC1, as the antibody used was raised against human PC1. Using the tandem fluorescent reporter mRFP-GFP-LC3, WT and PC1KO mIMCDs displayed both yellow (merged red and green) punctae (autophagosomes) and red punctae (autolysosomes), while the addition of Baf A1 led to a concentration of only yellow punctae (Figure 2F), which indicates that the autophagic flux is normal in PC1KO mIMCDs. Since canonical signaling players such as mTOR or AMPK were not significantly different between WT and PC1KO (Appendix A), we investigated transcriptional mechanisms. Next to LC3 mRNA, expression of other autophagy-related mRNAs (Atg5, Atg12, Becn1) was enhanced in PC1KO cells (Figure 2G). This was further associated with an upregulation of FoxO1, a well-known transcription factor of autophagy genes (Figure 2H). These data suggest that the elevation of basal autophagy in PC1KO mIMCDs is partially due to enhanced mRNA expression of autophagy genes, possibly related to enhanced FoxO1 activity.

### 2.3. Enhanced Starvation-Induced Autophagic Response in PC1KO mIMCDs

Both in WT and PC1KO mIMCDs, an upregulation of autophagy was evident following 3 h incubation with HBSS, as assessed with LC3 Western blotting (Appendix A), and following transfection with GFP-LC3 (Appendix A). However, cell death differences between WT and PC1KO were most clear following several days of starvation. Because autophagy activation typically precedes cell death initiation, we evaluate cell death following 72 h and autophagy following 48 h of incubation in HBSS throughout the remainder of the manuscript. Autophagy marker LC3-II was significantly increased in PC1KO mIMCDs compared to WT following 48 h of starvation, but this was only evident in the presence of Baf A1 (Figure 3A), suggesting an enhanced autophagic flux in PC1KO cells. While Sqstm1 levels were similar between the cells (Figure 3B), there was enhanced Sqstm1 mRNA transcription in PC1KO versus WT cells (Figure 3C), suggesting that a possible enhanced Sqstm1 degradation is compensated by upregulated expression. Similar results were obtained in GFP-LC3-transfected mIMCDs (Figure 3D). Introduction of hPC1 in PC1KO reduced LC3-II to levels more comparable to those of WT mIMCDs (Figure 3E), suggesting that this is directly dependent on PC1. Inhibition of autophagy by lysosomal inhibitor Baf A1 or by siRNA-mediated knockdown of autophagy protein BECN1 (siBECN1) during 48 h of HBSS in PC1KO cells reduced the number of live cells to levels comparable to Baf-A1- or siBECN1-treated WT mIMCDs (Figure 3F,G), supporting the hypothesis that reduced starvation-induced cell death in PC1KO mIMCDs is due to enhanced autophagy.

Unlike the enhanced basal autophagy in PC1KO cells, mRNA expression of autophagy-related genes was not significantly altered between WT and PC1KO cells, except for Atg12 (Figure 3H). Furthermore, FoxO1 was also similar between the cell lines (Figure 3I). Activity of autophagy-signaling players mTOR and AMPK was also not significantly different (Appendix A). However, the mRNA expression of stress-induced transcription factor Hif1α was significantly upregulated in PC1KO mIMCDs (Figure 3J).

### 2.4. Enhanced Autophagy in PC1KO Is Independent of PC2

Since stress-induced autophagy is proposed to be dependent on PC2, we investigated PC2 levels in PC1KO mIMCDs. First, although the protein levels decreased during starvation in both WT and PC1KO mIMCDs (Figure 4A), mRNA expression levels remained unaltered (Figure 4B). Nonetheless, significantly upregulated PC2 protein and Pkd2 mRNA levels in PC1KO mIMCDs compared to WT before and following starvation were observed (Figure 4A,B). These data suggest that upregulated PC2 might be causing enhanced autophagy in PC1KO mIMCDs. In addition, transient overexpression of hPC1 in PC1KO mIMCDs reduced the levels of PC2 towards WT levels (Figure 4C). We therefore performed siRNA-mediated knockdown of PC2 (siPC2) in WT and PC1KO mIMCDs subjected to starvation. Interestingly, autophagy levels following 48 h of starvation were not significantly altered upon knockdown of PC2 in PC1KO cells (Figure 4D). Cell death too remained unaltered following 72 h of starvation in siPC2-treated PC1KO mIMCDs and WT mIMCDs (Figure 4E).

### 2.5. Reduced Early but Enhanced Late Starvation-Induced Autophagic Response in PC2KO IMCDs

To further investigate the interplay between PC1 and PC2 during nutrient starvation in mIMCDs, we evaluated the autophagic response in PC2KO mIMCDs. Similar to the PC1KO, some residual PC2 mRNA was detected in PC2KO mIMCDs (Figure 5A), likely because the qPCR primers bind upstream (exon 1–2) of the site targeted by CRISPR-Cas9 (exon 2–3). On protein level, no PC2 could be detected (Figure 5B). No differences between WT, PC1KO or PC2KO were observed concerning morphology, proliferation or ATP production (Appendix A). In line with the literature, PC2KO IMCDs displayed lower basal LC3-II levels compared to WT or PC1KO cells, associated with a lack of early autophagic response following 3 h of nutrient starvation (Appendix A). However, following 48 h of starvation, PC2KO mIMCDs displayed enhanced autophagy compared to WT, as analyzed with LC3-II Western blotting (Figure 5B) and GFP-LC3 punctae measurements (Figure 5C). Interestingly, similar to the PC1KOs, LC3-II increase was only observed in the presence of Baf A1, suggesting enhanced autophagic flux (Figure 5B). This was associated with a slightly higher resistance to cell death in PC2KO mIMCDs following 72 h of starvation (Figure 5D) and significantly reduced Caspase 3 activation (Figure 5E). Similar to PC1KO mIMCDs, this was also associated with enhanced Hif1α mRNA expression (Figure 5F).

Because the late autophagic response following 48 h of nutrient starvation was enhanced in PC2KO mIMCDs but not in PC1KO mIMCDs treated with PC2 siRNA, we evaluated whether this could be caused by changes in PC1 expression. First, basal levels of PC1 were significantly increased in PC2KO mIMCDs compared to WT (Figure 6A), which was reflected in mRNA levels of Pkd1 (Figure 6B). Following starvation, however, while Pkd1 mRNA expression increased in both WT and PC2KO mIMCDs (Figure 6B), PC1 protein levels were only enhanced in WT cells, not in PC2KO, leading to significantly reduced PC1 in PC2KO compared to WT mIMCDs (Figure 6A). Moreover, the detected PC1 band in PC2KO seemed to be of a lower molecular weight (Figure 6A), suggesting that PC1 protein stability during starvation is dependent on PC2 expression. To test this, we aimed to suppress PC1 degradation by inhibition of either autophagic/lysosomal degradation (with Baf A1) or proteasomal degradation (with MG132). Interestingly, Baf A1 seemed to partially restore the PC1 levels in PC2KO cells subjected to 72 h of starvation (Figure 6C), suggesting that PC1 is degraded in PC2KOs by autophagy. In contrast, MG132 further increased PC1 degradation in both WT and PC2KO cells (Figure 6C), possibly due to enhanced autophagy compensating for the loss of proteasomal degradation. Restoring the PC1 levels by overexpression of hPC1 in PC2KO reverted LC3-II levels (Figure 6D) and cell death (Figure 6E) following starvation back to levels comparable to WT mIMCDs. These data support the hypothesis that the findings observed in PC2KO are dependent on the reduced levels of PC1.

### 2.6. PTECs from ADPKD Patients Are More Resistant towards Nutrient Starvation

In order to establish that our findings are relevant for ADPKD, we evaluated key findings of this study in conditionally immortalized proximal tubular epithelial cells (PTECs) derived from urine of young ADPKD patients with heterozygous frameshift *PKD1* mutations and young healthy individuals (Appendix A). Cells from ADPKD patients had ca. 50% lower PC1 levels compared to healthy controls (Figure 7A,B). Unlike PC1KO mIMCDs, ADPKD PTECs did not display increased PC2 levels (Figure 7A,B).

Basal autophagy, as indicated by LC3-II levels, was not significantly different in ADPKD versus control PTECs both in the absence and presence of Baf A1, despite higher levels of precursor LC3-I (Figure 7C). The increase of LC3-I was associated with higher LC3 mRNA expression in ADPKD PTECs (Figure 7D), similar to the findings in IMCDs. This was also associated with significantly higher FoxO1 levels (Figure 7E). Following starvation, LC3-II levels were higher in the ADPKD compared to control PTECs (Figure 7F), which was associated with less reduction in confluence (Figure 7G) and higher cell viability (Figure 7H). As such, enhanced starvation-induced autophagy and resistance against nutrient starvation are already prominent in early-stage ADPKD cells with reduced PC1 levels.

## 3. Discussion

The polycystins are involved in, among others, cellular Ca^2+^ signaling, cAMP signaling, cell proliferation, apoptosis, metabolism, polarity, adhesion, inflammation and autophagy [12]. However, the exact dysregulation of autophagy in ADPKD remains a matter of debate. Enhanced autophagy was observed in late-stage cpk and Han:SPRD cystic kidneys [13], while pkd1a^−/−^ zebrafish, mouse kidney epithelial cells of Pkd1^−/−^ mice and cyst-lining cells displayed impaired autophagic flux [14]. Moreover, in Pkd1 miRNA transgenic mice, mRNA expression of autophagy genes was reduced, despite no changes in LC3-II protein level compared to WT littermates [23]. In one immortalized cystic cell line (OX161) versus one normal renal epithelial cell line (UCL93), autophagy was also reduced [24]. Suppressed autophagic flux was observed in the heart of a hypomorphic Pkd1 mouse model [25] and decreased in the kidney [15]. However, microarray data of ADPKD patient samples did not reveal reduced autophagy, while LC3 was strongly upregulated in cystic tissues of some ADPKD patients [26]. Concerning PC2, findings are more consistent: PC2 seems to be indispensable for stress-induced autophagy [16,17,18,19].

Here we demonstrated reduced starvation-induced cell death in polycystin-deficient mouse and human ADPKD cells concurrent with increased autophagic flux in comparison to WT or healthy cells. During longer periods of starvation, generally a transition occurs where autophagy is downregulated and cell death increases [27]. Our data imply that polycystins regulate this transition from survival to death, which is around 48–72 h of starvation in mIMCDs. This transition seems to be fine-tuned by the correlated expression of PC1 and PC2 proteins, where PC1 regulates PC2′s levels and vice versa. Despite enhanced levels of PC2 in PC1KO, caused by enhanced transcription, knockdown of PC2 did not alter autophagy levels in PC1KO mIMCDs. During starvation, PC1 levels are clearly elevated in WT, but not in PC2KO mIMCDs, which is partially caused by the enhanced autophagic/lysosomal degradation observed in PC2KO cells. This is an interesting finding as PC1 is described as being regulated by proteasomal degradation [28]. Proteasome inhibition, however, further reduced PC1 levels in WT and PC2KO mIMCDs. This can be attributed to compensatory autophagy stimulation caused by proteasome inhibition [29]. Eventually, enhanced autophagy and reduced cell death in PC2KO mIMCDs were restored by overexpression of hPC1. This implies that PC1 is more directly involved in stimulating the transition from autophagy to cell death, rather than PC2. However, PC2 can still regulate autophagy and cell death through stabilizing PC1 levels.

Mechanistically, we tested the expression of different regulators of basal or starvation-induced autophagy. Enhanced basal autophagy (observed in PC1KO, but not PC2KO mIMCDs) was associated with higher FoxO1 levels and enhanced mRNA expression of various autophagy genes. FoxO transcription factors, mostly FoxO1 and FoxO3, are well-known mediators of autophagy, and our data now suggest that their levels are regulated by PC1. Interestingly, FoxO1 was identified as one of the key genes involved in ADPKD progression derived from microarray dataset analysis [30]. Whether FoxO suppression attenuates cyst development and whether this is dependent on autophagy remains to be investigated.

Although FoxO1 stimulation could prime the cells for autophagy stimulation, enhanced FoxO1 levels and associated transcription were not observed in PC1KO mIMCDs following 48 h of starvation. Moreover, PC2KO (which did not show enhanced basal autophagy) also displayed enhanced starvation-induced autophagy, suggesting that another mechanism contributes to enhanced autophagy following starvation. In this regard, stress-induced transcription factor Hif1α has been implicated in cyst development and enhanced autophagy in a rat model of polycystic kidney disease [13]. In our hands, enhanced Hif1α mRNA expression was also observed in PC1KO and PC2KO cells following starvation. Whether Hif1α mRNA expression also reflects that its protein levels are increased, and whether this is merely associated with enhanced starvation-induced autophagy in these cells, obviously requires further investigation. Interestingly, Hif1α deletion attenuated cyst progression in an ADPKD mouse model [31].

It should be noted that studies investigating autophagy are often limited to animal models or human cellular models comparing healthy kidney cells and cyst-lining cells [32]. These represent late-stage ADPKD cells that have undergone drastic morphological and molecular changes, partially in response to the stressed cystic environment. As such, altered autophagy in late-stage cystic cells could represent a secondary effect rather than a direct effect from polycystin loss (early stage) [12]. In this respect, it is interesting to note that kidney lysates of 70-day-old hypomorphic Pkd1 mice displayed enhanced levels of autophagy markers, while most of these markers were reduced in 120-day-old mice [15]. Here the polycystin-deficient mIMCDs lacked enhanced mTOR activity characteristic of cystic cells, while the PTECs were derived from young early-stage ADPKD patients that carry one mutant *PKD1* allele.

Cystogenesis is proposed to be triggered by a second-hit mechanism and/or gene dosage effects [33,34]. The gene dosage hypothesis suggests that the dosage of functional PC1 or PC2 protein determines cystogenesis rate [35]. An additional “third hit”, identified as renal ischemia-reperfusion [36] or toxic injury [37], was proposed to enhance cyst formation in ADPKD [38]. Our data now reconcile the gene dosage and third hit hypotheses: resistance against stress is inversely proportional to polycystin (most prominently PC1) levels and cells with the lowest levels of PC1 will endure nutritional stress better. This is reflected in the observation that LC3 overexpression in an Ift46^−/−^ PKD model enhanced cyst formation [26]. Moreover, renal stress is inherent to the ADPKD phenotype: cyst expansion leads to mechanical stress to the surrounding tissue, activating injury-related pathways and inducing a snowball effect triggering the formation of new cysts [39,40]. Whether this inverse relation of polycystin levels with survival also involves other stress conditions remains to be investigated.

There are several limitations of our study. First, although our findings provide some insight into the molecular mechanisms behind cyst progression in ADPKD, translation towards clinical management in ADPKD is difficult. In general, our data provide fundamental evidence that an increased autophagic flux provides a survival benefit for cells carrying mutations in PC1 and PC2, suggesting that these cells would be especially sensitive to suppression of autophagy. However, whether this would provide a sensible therapeutic strategy in a living organism is far from clear. However, our study adds to the specific characterization of ADPKD epithelial cells, and contributes to the delineation of cell-specific properties that could be exploited for future therapeutic purposes. Further research is obviously required to further explore this. Second, the data in our study were mostly produced from experiments in WT, PC1KO and PC2KO mIMCDs. One should consider the limits of generating and working with cell lines. The PC2KO mIMCDs have been described before [41], and the PC1KO cells were generated simultaneously in a similar manner. Key findings regarding autophagy and cell death were reversed when we overexpressed PC1 and were validated in human ADPKD cell lines with *PKD1* mutations, suggesting that the findings in mIMCDs are not the result of side effects during cell line generation.

In conclusion, polycystin-deficient cells are more resistant against nutrient stress by delaying the transition from cell survival to cell death. This is achieved by sustaining autophagy, modulated by PC1 in a PC2-dependent manner. This could have strong implications for cystogenesis, as the renal cells with the lowest PC1 levels, which are also most prone to form cysts, will survive and recover better from a lack of nutrients.

## 4. Materials and Methods

### 4.1. Cell Lines, Starvation and Transfection

Mouse inner medullary collecting duct cells mIMCD-3 (ATCC CRL-2123) were grown in DMEM:F-12 (Biowest) with 10% fetal bovine serum (FBS; Westburg) and 1% Pen/Strep (P/S; Westburg). PC2 knockout (PC2KO) mIMCDs were described previously [41]. PC1 knockout (PC1KO) mIMCDs were generated in a similar fashion (Appendix A). Pkd1 single guide RNA (sgRNAs) sequences near exon 2 and exon 3 of the mouse *Pkd1* genomic DNA were designed with the CRISPR Design Tool (http://crispr.mit.edu/, accessed on 15 January 2017). sgRNAs 5′-GATAGTCAGGAGACGAGCTC and 5′-CTACTTCAGACGCTGGACAT were used for exon 2 and 5′-CCCTGCAAGAAAGAAAGATG and 5′-AGGATTTCTACATTAGAAGA for exon 3 editing. Two pairs of 20-nucleotide sgRNAs were cloned into pGL3-U6-sgRNA-PGK-Hygromycin (modified from pGL3-U6-sgRNA-PGK-puromycin; Addgene 51133). Plasmids CMV-hspCas9 (D10A)-T2A-Puro (CASLV100PA-1, System Biosciences, Inc, Palo Alto, CA, USA), pMDLg/pRRE, pRSV-Rev and pMD2.G were transfected into human embryonic kidney (HEK) 293T cells with Lipofectamine 2000 (Invitrogen, Waltham, MA, USA) to generate Cas9-D10A-LV. Cas9-D10A-LV was infected into mIMCD3 cell lines and selected with puromycin to obtain the stable Cas9-D10A mIMCD3 wild-type (WT) cell lines. pGL3-U6-sgRNA-PGK-Hygromycin with *Pkd1*-specific sgRNAs was then transfected into WT mIMCD3 stable cells. Individual cells were selected with hygromycin and puromycin. Genomic DNA was extracted from the single clones of the targeted mIMCD3 cells and PCR amplification of the genomic DNA was performed using the primer set flanking the exon 2 and exon 3. The primer sequences are Forward primer: 5′-TGTGTCTCAGTGCCTGCCACTGAAG and Reverse primer: 5′-TGATGTTCCTTTGCCCAGCGTGGCAG. PCR products were analyzed by electrophoresis followed by direct DNA sequencing of the gel-purified products using the Forward primer. PC1KO mIMCD3 cell lines carrying frameshift insertions-deletions (indels) in exon 2 and exon 3 of *PKD1* were selected. The stable Cas9-D10A mIMCD3 WT cell lines were used as control.

Proximal tubular epithelial cells (PTECs) were collected from the urine of healthy individuals or genotyped ADPKD patients with a heterozygous frameshift mutation in PKD1, immortalized and subcloned as previously described [42]. Informed consent was obtained from all participants and all study procedures were approved by the ethical committee of UZ Leuven (S51837). In short, a pellet from centrifuged urine was grown in supplemented DMEM:F-12. Primary PTECs are then conditionally immortalized with a retroviral construct containing SV40_LT_tsA58. This temperature-sensitive SV40 large T (LT) antigen makes SV40 LT unstable at 37 °C [43]. Hence, conditionally immortalized PTECs are cultured at 33 °C. For experiments, monoclonal PTECs are incubated for 10 days at 37 °C, which initiates cell differentiation [44,45]. All cell lines are frequently tested for mycoplasma contamination.

For starvation experiments, medium is gently replaced with Hanks’ balanced salt solution (HBSS) containing calcium and magnesium (Gibco, Thermo Fischer Scientific, Waltham, MA, USA).

To measure the autophagic flux, mIMCDs were treated with 100 nM of lysosomal blocker Bafilomycin A1 (Baf A1; B-1080, LC Laboratories, Woburn, MA, USA) or DMSO as control 3 h before cell collection. The addition of Baf A1 blocks the lysosomal degradation of autophagosomal marker LC3-II and allows us to examine the production rate of novel autophagosomes [46].

To analyze the PC1 degradation during starvation, mIMCDs were subjected to HBSS for 72 h with the last 24 h in the presence of 100 nM Baf A1 or 1 µM proteasomal inhibitor MG132 (Wako Chemicals 139-18451, Neuss, Germany).

For transfection, plasmids are mixed with JetPrime reagent and buffer (Polyplus Transfections, Illkirch-Graffenstaden, France) according to the manufacturer’s protocol. Reagent versus plasmid ratio was 2:1.

For siRNA-mediated knockdown, 100 nM of siRNA was incubated with Dharmafect (Dharmacon, Horizon Discovery, Waterbeach, UK) according to the manufacturer’s protocol. The next day, the medium was replaced with normal medium or HBSS.

### 4.2. Plasmids, siRNA and Antibodies

The following plasmids were used for transfection: pcDNA3.1(-)-GFP-LC3 [47], mRFP-GFP-LC3 construct [48]. hPC1-expressing plasmid WT (REP10) was a gift from Gregory Germino (Addgene plasmid #21368) [49] as was hPC2-expressing plasmid (M-PKD2 (OF2-3); Addgene plasmid #21370) [21].

The following siRNAs were used: siRNA against mouse Pkd2 (AM16708, assay ID 150154, Thermo Fischer Scientific), siRNA against mouse BECN1 (4390771, Ambion, Woolston, UK) and negative control DsiRNA (51-01-14-04, IDT).

The following antibodies were purchased from Cell Signaling Technology (Danvers, MA, USA): anti-β-Actin (4970), anti-FoxO1 (14952), anti-ULK1 (8054), anti-phospho-ULK1 (Ser555) (5869), anti-phospho-ULK1 (Ser757) (14202), anti-phospho-ULK1 (Ser317) (3776), anti-AMPK (5832), anti-phospho AMPK (Thr172) (50081), anti-S6 (2217) and anti-p-S6 (4858). Anti-PC1 (sc-130554), anti-PC2 (sc-28331) and anti-SV40 LT were from Santa Cruz Biotechnology (Dallas, TX, USA), anti-Sqstm1 (P0067) and anti-EpCam (HPA026761) from Sigma-Aldrich (Saint-Louis, MO, USA, anti-LC3 from Nanotools (clone 5F10; München, Germany), anti-PgP from Abcam (ab170904; Cambridge, UK) and anti-AQP1 from Novus Biologicals (84488; Abingdon, UK). Secondary anti-rabbit and anti-mouse antibodies were from Bio-Rad Laboratories (Hercules, CA, USA).

### 4.3. Quantitative PCR (qPCR)

RNA is extracted from cell pellets using the RNeasy Mini Kit (Qiagen, Hilden, Germany). Complementary cDNA is then synthesized using the High-Capacity cDNA Reverse Transcription Kit (Applied Biosystems, Waltham, MA, USA). qPCR is performed using the Applied Biosystems StepOnePlus Real-Time PCR systems (Thermo Fischer Scientific). Primer sets were validated with transient temperature PCR and Platinum SYBR Green qPCR Supermix-UDG (Thermo Fischer Scientific). Following primer pairs were selected for LC3 (MAP1LC3B) mRNA: 5′-GCGACTGGAGAGCTGTTTCT and 5′-AACCACATCCTAAGGCCAGC; for beta-ACTIN: 5′-ATGCTCCCCGGGCTGTAT and 5′-CATAGGAGTCCTTCTGACCATTC; and for GAPDH: 5′-TGTGTCCGTCGTGGATCTGA and 5′-CCTGCTTCACCACCTTCTTGA. (Taqman-probes were then created for LC3, beta-ACTIN (5′-CCTAGGCACCAGGGTGTGATG) and GAPDH (5′-CCGCCTGGAGAAACCTGCCAAGTATG) and combined with the PrimeTime Gene Expression Master Mix from Thermo Fischer Scientific. For other mRNA, we used primer and probe sets from Thermo Fischer Scientific: Mm00465434_m1 (mPKD1; spanning exon 1–2), Mm00435829_m1 (mPKD2; spanning exon 1–2), Mm01187303_m1 (mATG5), Mm00503201_m1 (mATG12), Mm01265461_m1 (mBECN1), Mm00448091_m1 (mSQSTM1) and mHif1a (Mm00468869_m1). qPCR was performed with the corresponding StepOnePlus Real-Time PCR System Software and analyzed with the 2^−ΔΔCt^ method [50].

### 4.4. Western Blotting

Cells were homogenized in a RIPA lysis buffer containing 10 mM sodium phosphate (pH 7.5), 150 mM NaCl, 1.5 mM MgCl2, 0.5 mM DTT, 1% Triton X-100, protease and phosphatase inhibitors (Roche). Protein concentration was determined by bicinchoninic acid assay (Thermo Fischer Scientific). Equal µg of samples was prepared with SDS buffer containing β-mercaptoethanol, heated at 95 °C for 3 min and loaded on NuPAGE 4–12% Bis-Tris gels (Thermo Fischer Scientific). SDS-PAGE was performed with a constant voltage of 160 V. Next, proteins were blotted on polyvinylidene difluoride (PVDF) membranes using the Mini Blot module set (Thermo Fischer Scientific) for 2 h at 30 V. Membranes were blocked for 1 h at room temperature with PBS-Tween (0.1%) containing 5% milk powder (MP) followed by incubation with the primary antibody diluted in PBS-T and 2% MP overnight at 4 °C. Next day, the membranes were washed 3 times with PBS-T and then incubated with the secondary horseradish peroxidase (HRP)-coupled antibody for 45 min with PBS-T + 2% MP. After washing 3 times with PBS-T, immunoreactive bands were visualized through enhanced chemoluminescence (Pierce ECL Plus Western Blotting Substrate), followed by serial exposure and detection with the G:Box Gel Doc system from Syngene (Cambridge, UK). Intensity quantification was performed on unsaturated bands with ImageJ [51].

### 4.5. (mRFP)-GFP-LC3 Measurements

mIMCDs transfected with pcDNA3.1(-)-GFP-LC3 or mRFP-GFP-LC3 construct were fixed in 4% paraformaldehyde 72 h after transfection and washed with PBS. mRFP-GFP-LC3 allows us to distinguish early autophagosomes (exciting RFP and GFP) from autolysosomes (exciting only RFP) [48]. A mixture of moviol and DAPI (0.5 μg/mL) was added per chamber. Cells were acquired using a fluorescence microscope (Nikon Eclipse Ci) with UV laser, Nikon’s DS-Fi3 camera and the NIS-Elements Microscope Imaging Software. The appropriate filters were used for DAPI (Exc: 340–380 nm; Em: 435–485 nm), GFP (Exc: 465–495 nm; Em: 515–555 nm) and RFP (Exc: 528–553 nm; Em: 590–650 nm). Per experiment, 15–20 random images were taken and afterwards randomized with a number. The amount of punctae per cell was quantified in a blinded fashion by 2 persons independently.

### 4.6. IncuCyte Live-Cell Imaging

Cells were seeded in 6-well plates and incubated in the IncuCyte S3 Live-Cell analysis system (Sartorius, Göttingen, Germany) to monitor cell growth. To measure cell death, Cytotox Green at a final concentration of 250 nM was added to the medium. Confluence and Cytotox Green signals were afterwards measured using the IncuCyte base analysis software. Cell number was quantified in unbiased fashion using the IncuCyte cell-by-cell analysis software module.

### 4.7. Trypan Blue Exclusion Assay

Following HBSS treatment, medium, PBS and trypsinized cells were pooled and 10 µL of suspension was mixed with 10 µL 0.4% (*w*/*v*) Trypan Blue. The number of live and dead cells was determined in duplicate using dual-chambered slides and TC20 automated cell counter (Bio-Rad).

### 4.8. ATP/ADP Measurements

ATP/ADP ratio was measured and quantified using the ADP/ATP ratio assay kit (MAK135; Sigma-Aldrich, Saint-Louis, MA, USA) according to the manufacturer’s instructions.

### 4.9. Statistical Analysis

Statistical analysis was performed using GraphPad Prism. Data are expressed as mean ± standard deviation. Number of independent experiments (N) is annotated in figure legends. After evaluation of normal distribution (Shapiro–Wilk normality test) and equal variances (F-test), a Student’s *t*-test was performed to compare 2 groups, while ANOVA was used for comparison of more than 2 groups. In the experiments in mIMCDs, the observations of each experiment were considered as paired or repeated measurements. For experiments in human PTECs, unpaired *t*-test or nested *t*-test was used. To evaluate the effects of both cell line and a specific treatment, we used two-way ANOVA.

## Figures and Tables

**Figure 1 ijms-22-13511-f001:**
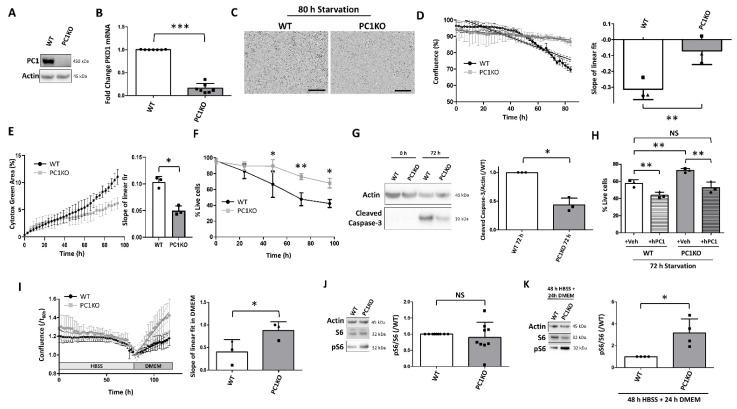
Less starvation-induced cell death in PC1KO mIMCDs. (**A**) Knockout of polycystin-1 (PC1KO) in mouse inner medullary collecting duct cells (mIMCDs) was confirmed by Western blot. (**B**) Reduced Pkd1 mRNA expression was confirmed by quantitative PCR (qPCR) (N = 7). (**C**) Representative microscopic brightfield images of wild-type (WT) and polycystin-1 knockout (PC1KO) mouse inner medullar collecting duct cells (mIMCDs) after 80 h of starvation by incubation in Hank’s balanced salt solution (HBSS). WT mIMCDs are less confluent and show more dark rounded detached cells. Scale bar = 300 µm. (**D**) Analysis of confluence of WT (black) and PC1KO (grey) mIMCDs during incubation in HBSS using the IncuCyte Live Cell Analyzer. Left: quantification of % confluence over time in HBSS with linear fit and 95% confidence interval (dashed line) superimposed; right: quantification of the slope of the linear fit of the confluence over time. Paired observations of each independent experiment are represented by the same symbol (N = 3). (**E**) Analysis of Cytotox Green signal increase during incubation in HBSS in WT (black) and PC1KO (grey) mIMCDs. Left: quantification of % of Cytotox Green signals over the total area of the image over time in HBSS; right: quantification of the slope of the linear fit over time (N = 3). (**F**) Percentage live cells over time in HBSS as analyzed by Trypan Blue exclusion in WT (black) and PC1KO (grey) mIMCDs (N = 4). (**G**) Apoptosis was analyzed by Western blotting of cleaved Caspase 3 in WT and PC1KO mIMCDs. Left: Representative Western blot of protein lysates following 0 h or 72 h of nutrient starvation; right: quantification of cleaved Caspase 3 levels over Actin (N = 3). (**H**) Percentage live cells following 72 h of incubation in HBSS as analyzed by Trypan Blue exclusion in WT and PC1KO mIMCDs, transfected either with Vehicle (+Veh) or human PC1 (+hPC1). Paired observations of each independent experiment are represented by the same symbol (N = 3). (**I**) Analysis of confluence of WT (black) and PC1KO (grey) mIMCDs during incubation in HBSS, followed by recovery in normal medium (DMEM) using the IncuCyte Live Cell Analyzer. Left: quantification of confluence (normalized to the initial time of recovery), with the linear fit in DMEM superimposed. The dashed line is the 95% interval of the linear fit; right: quantification of the slope of the linear fit in DMEM during recovery. Paired observations of each independent experiment are represented by the same symbol (N = 3). (**J**) Analysis of phosphorylated S6 (pS6) levels in WT and PC1KO mIMCDs in basal conditions. Left: representative Western blot; Right: quantification of pS6 levels over total S6 (N = 9); (**K**) Analysis of pS6 levels in WT and PC1KO mIMCDs following 48 h of HBSS and 24 h of recovery in DMEM. Left: representative Western blot; Right: quantification of pS6 levels over total S6 (N = 4). NS: not significant, ^∗^
*p* < 0.05, ^∗∗^
*p* < 0.01, ^∗∗∗^
*p* < 0.001.

**Figure 2 ijms-22-13511-f002:**
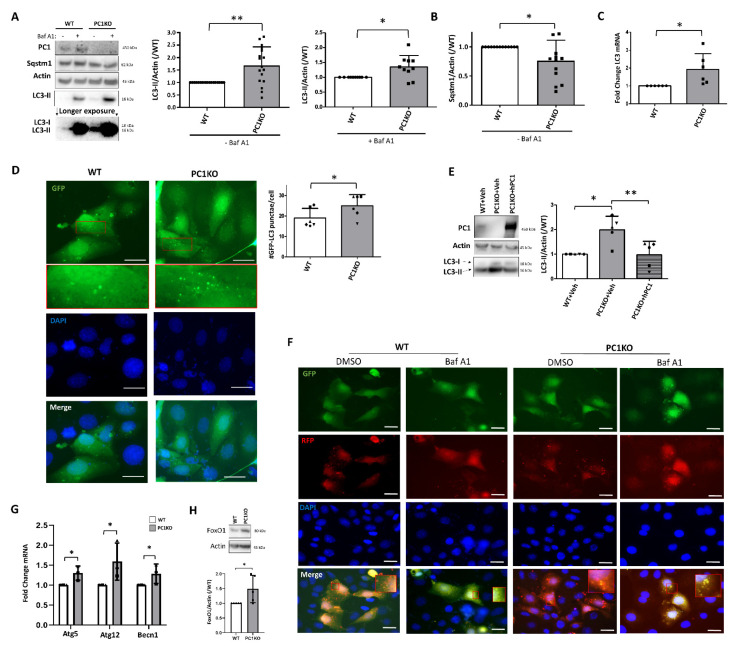
Increased basal autophagy in PC1KO mIMCDs. (**A**) Basal autophagy levels were assessed by Western blotting for LC3 and Sqstm1. Wild-type (WT) and PC1KO mIMCDs were incubated for 24 h in fresh medium and 100 nM Bafilomycin A1 (Baf A1; +) or DMSO (−) was added 3 h prior to harvest. Left: representative Western blot; middle: quantification of LC3-II levels over Actin in absence of Baf A1 (N = 16); right: quantification of LC3-II levels over Actin in presence of Baf A1 (N = 10). (**B**) Quantification of Sqstm1 levels over Actin in absence of Baf A1 (N = 12). (**C**) qPCR for LC3 mRNA in WT and PC1KO mIMCDs (N = 6). (**D**) Representative GFP, DAPI, and merged images of GFP-LC3-transfected WT and PC1KO mIMCDs (scale bar = 20 µm). The insets show the GFP-LC3 punctae. Right: quantification of number of GFP-LC3 punctae per cell. Paired observations of each independent experiment are represented by the same symbol (N = 6). (**E**) Overexpression of human PC1 (hPC1) in PC1KO restored LC3-II levels. Left: representative Western blot of protein lysates of WT or PC1KO IMCDs transfected with Vehicle (+Veh) or human PC1 (+hPC1); right: quantification of LC3-II over Actin. Paired observations of each independent experiment are represented by the same symbol (N = 5). (**F**) Analysis of mRFP-GFP-LC3 punctae in transiently transfected WT and PC1KO mIMCDs. Shown are representative GFP, RFP, DAPI and merged images (scale bar = 20 µm) (N = 3). Following addition of Baf A1 (100 nM, 3 h), only combined green and red punctae were observed in WT and PC1KO mIMCDs. (**G**) qPCR for Atg5, Atg12 and Becn1 mRNA in WT (white) and PC1KO (grey) mIMCDs (N = 3). (**H**) FoxO1 protein levels in WT and PC1KO mIMCDs. Upper: representative Western blot; lower: quantification of FoxO1 levels over Actin (N = 5). ^∗^
*p* < 0.05, ^∗∗^
*p* < 0.01.

**Figure 3 ijms-22-13511-f003:**
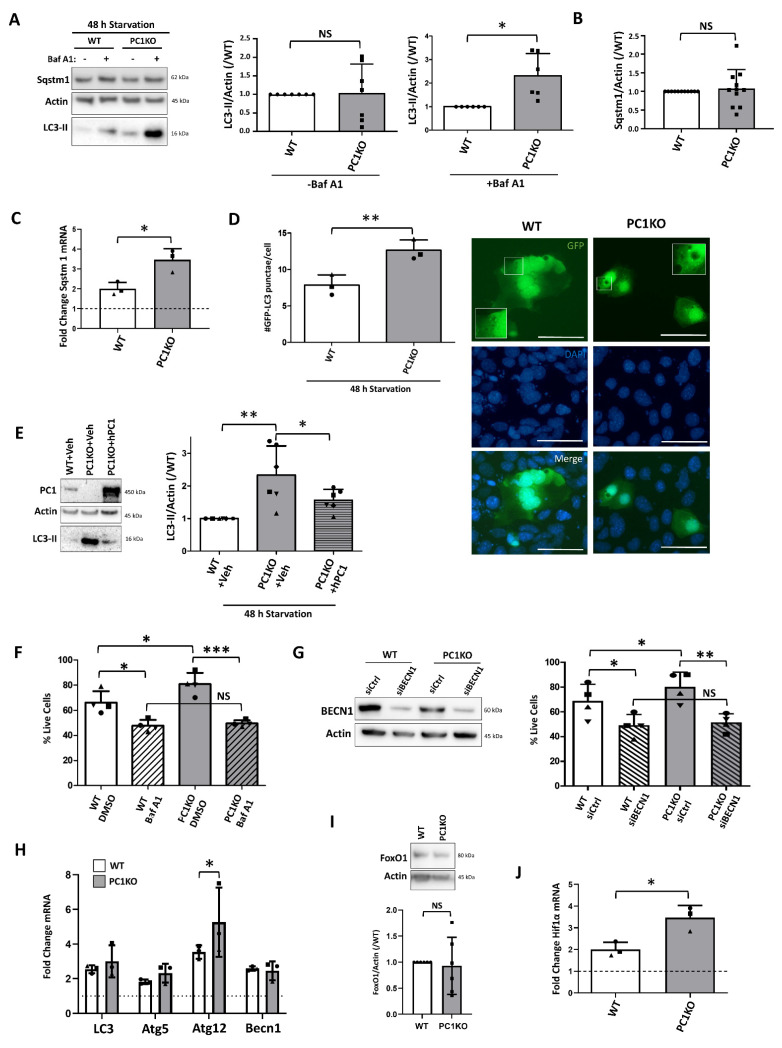
Increased autophagy in PC1KO mIMCDs following 48 h of starvation leads to enhanced resistance against cell death. (**A**) LC3-II and Sqstm1 levels were analyzed in protein lysates from cells subjected to 48 h of nutrient starvation. Three hours before harvest, DMSO (- Baf A1) or Bafilomycin A1 (100 nM; + Baf A1) was added. Left: representative Western blot; middle: analysis of LC3-II levels over Actin in absence of Baf A1; right: analysis of LC3-II levels over Actin in presence of Baf A1 (N = 7). (**B**) Quantification of Sqstm1 levels over Actin following 48 h of starvation in absence of Baf A1 (N = 11). (**C**) qPCR for Sqstm1 mRNA in WT and PC1KO mIMCDs subjected to 48 h of starvation. The dashed line represents the WT expression in basal conditions (N = 3). (**D**) GFP-LC3 punctae analysis in WT and PC1KO mIMCDs following 48 h of starvation. Left: representative GFP, DAPI and merged images (scale bar = 25 µm); right: quantification of the number of GFP-LC3 punctae per cell following 48 h of starvation. Paired observations of each independent experiment are represented by the same symbol (N =3). (**E**) Overexpression of human PC1 (hPC1) in PC1KO restored LC3-II levels. Left: representative Western blot of protein lysates of WT or PC1KO IMCDs transfected with Vehicle (+Veh) or human PC1 (+hPC1); right: quantification of LC3-II over Actin. Paired observations of each independent experiment are represented by the same symbol (N = 6). (**F**) Percentage live cells following 48 h of incubation in HBSS as analyzed by Trypan Blue exclusion in WT and PC1KO mIMCDs, treated either with DMSO or 100 nM Baf A1. Paired observations of each independent experiment are represented by the same symbol (N = 4). (**G**) Percentage live cells following 48 h of incubation in HBSS as analyzed by Trypan Blue exclusion in WT and PC1KO mIMCDs, treated either with control siRNA (siCtrl) or siRNA against BECN1 (siBECN1). Left: representative Western blot showing the knockdown of BECN1; right: quantification of the % of live cells. Paired observations of each independent experiment are represented by the same symbol (N = 4). (**H**) qPCR for LC3, Atg5, Atg12 and Becn1 mRNA in WT (white) and PC1KO (grey) mIMCDs subjected to 48 h of starvation. The dashed line represents the WT expression in basal conditions (N = 3). (**I**) FoxO1 protein levels in WT and PC1KO mIMCDs subjected to 48 h of starvation. Upper: representative Western blot; lower: quantification of FoxO1 levels over Actin (N = 5). (**J**) qPCR for Hif1α mRNA in WT (white) and PC1KO (grey) mIMCDs subjected to 48 h of starvation. The dashed line represents the WT expression in basal conditions (N = 3). NS: not significant, ^∗^
*p* < 0.05, ^∗∗^
*p* < 0.01, ^∗∗∗^
*p* < 0.001.

**Figure 4 ijms-22-13511-f004:**
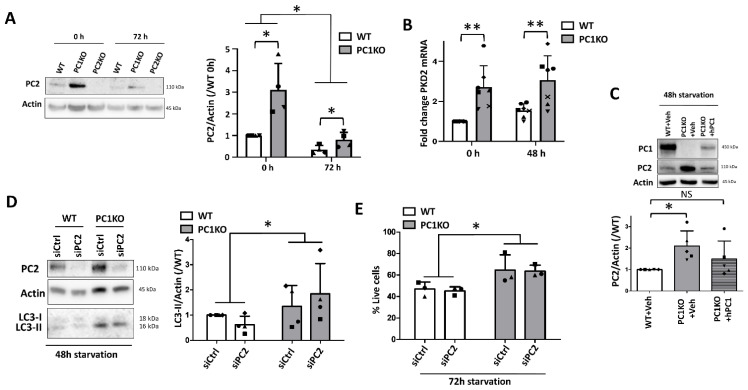
PC2 is upregulated in PC1KO, but is not involved in enhanced starvation-induced autophagy. (**A**) Western blot analysis of PC2 protein levels in WT, PC1KO and PC2KO IMCDs in full medium (0 h) or subjected to 72 h of nutrient starvation (72 h). Left: representative Western blot; right: quantification of PC2 levels over Actin in WT and PC1KO cells, normalized to WT levels (N = 4). (**B**) Pkd2 mRNA expression in basal conditions (0 h) or following 48 h of nutrient starvation (48 h) was evaluated by quantitative PCR (qPCR) (N = 7). (**C**) Western blot analysis of PC2 protein levels in WT or PC1KO IMCDs transfected with Vehicle (+Veh) or human PC1 (+hPC1). Upper: representative Western blot; lower: quantification of PC2 levels over Actin, normalized to WT levels. Paired observations of each independent experiment are represented by the same symbol (N = 5). (**D**) Western blot analysis of LC3-II in WT or PC1KO IMCDs, transfected with control (siCtrl) or anti-PC2 (siPC2) siRNA and subjected to 48 h of starvation. Left: representative Western blot; right: quantification of LC3-II over Actin, normalized to WT. Paired observations of each independent experiment are represented by the same symbol. Two-way ANOVA revealed significant differences between WT and PC1KO mIMCDs, but no effect of siPC2 (N = 4). (**E**) Percentage live cells following 72 h of starvation as analyzed by Trypan Blue exclusion in WT and PC1KO mIMCDs, transfected either with control (siCtrl) or anti-PC2 (siPC2) siRNA. Paired observations of each independent experiment are represented by the same symbol. Two-way ANOVA revealed a significant difference between WT and PC1KO mIMCDs, but no effect of siPC2 (N = 3). NS: not significant, ^∗^
*p* < 0.05, ^∗∗^
*p* < 0.01.

**Figure 5 ijms-22-13511-f005:**
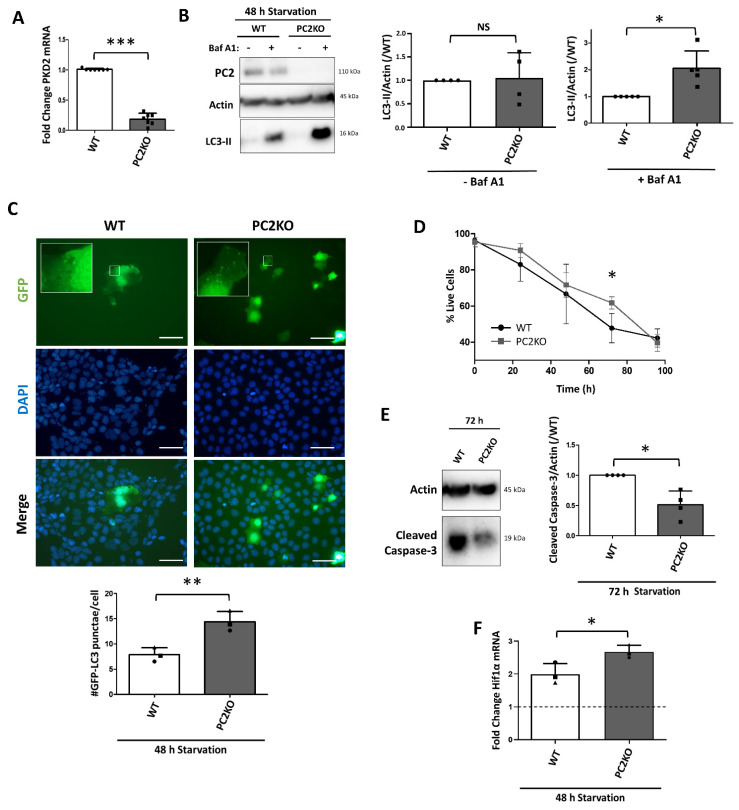
Increased starvation-induced autophagy and reduced cell death in PC2KO IMCDs. (**A**) Reduced Pkd2 mRNA expression in PC2KO mouse inner medullary collecting duct cells (mIMCDs) was confirmed by quantitative PCR (qPCR) (N = 7). (**B**) LC3-II levels were analyzed in protein lysates from cells subjected to 48 h of nutrient starvation. Three hours before harvest, DMSO (− Baf A1) or Bafilomycin A1 (100 nM; +Baf A1) was added. Left: representative Western blot; middle: quantification of LC3-II levels over Actin in absence of Baf A1 (N = 4); right: quantification of LC3-II levels over Actin in presence of Baf A1 (N = 5). (**C**) GFP-LC3 punctae analysis in WT and PC2KO mIMCDs following 48 h of starvation. Upper: representative GFP, DAPI and merged images (scale bar = 25 µm); lower: quantification of the number of GFP-LC3 punctae per cell following 48 h of starvation. Paired observations of each independent experiment are represented by the same symbol (N = 3). (**D**) Percentage live cells over time in HBSS as analyzed by Trypan Blue exclusion in WT (black) and PC2KO (dark grey) mIMCDs (N = 4). (**E**) Apoptosis was analyzed by Western blotting of cleaved Caspase 3 in WT and PC2KO mIMCDs. Left: representative Western blot of protein lysates following 0 h or 72 h of nutrient starvation; right: quantification of cleaved Caspase 3 levels over Actin (N = 4). (**F**) qPCR for Hif1α mRNA in WT (white) and PC2KO (grey) mIMCDs subjected to 48 h of starvation. The dashed line represents the WT expression in basal conditions (N = 3). NS: not significant, ^∗^
*p* < 0.05, ^∗∗^
*p* < 0.01, ^∗∗∗^
*p* < 0.001.

**Figure 6 ijms-22-13511-f006:**
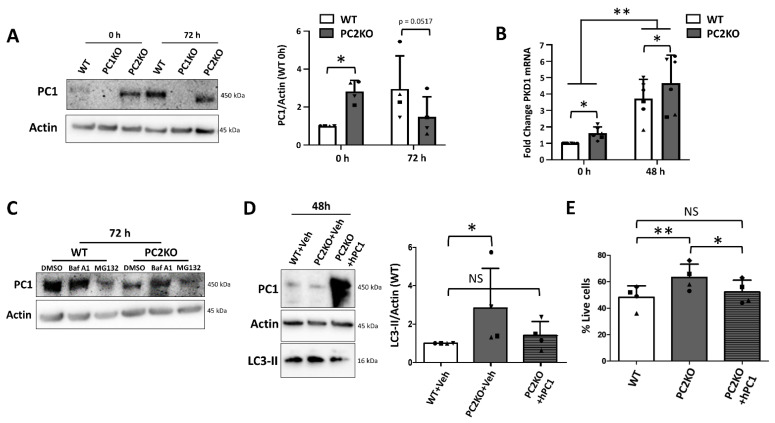
Altered regulation of PC1 expression in PC2KO mIMCDs is linked with enhanced autophagy. (**A**) Western blot analysis of PC1 protein levels in WT, PC1KO and PC2KO mIMCDs incubated in full medium (0 h) or subjected for 72 h to nutrient starvation (72 h). Left: representative Western blot; right: quantification of PC1 levels over Actin in WT and PC2KO cells, normalized to WT levels (N = 8). (**B**) Pkd1 mRNA expression in basal conditions (0 h) or following 48 h of nutrient starvation (48 h) was evaluated by quantitative PCR (qPCR). Two-way ANOVA revealed a significant difference between 0 h and 48 h, and between WT and PC2KO mIMCDs (N = 6). (**C**) Representative Western blot of PC1 in WT and PC2KO mIMCDs subjected to 72 h of nutrient starvation. In the final 24 h before harvest, 100 nM Bafilomycin A1 (Baf A1), 1 µM MG132 or DMSO was added. (N = 3). (**D**) Western blot analysis of LC3-II in WT or PC2KO mIMCDs transfected with Vehicle (+Veh) or human PC1 (+hPC1) followed by 48 h of starvation. Left: representative Western blot; right: quantification of LC3-II levels over actin, normalized to WT levels. Paired observations of each independent experiment are represented by the same symbol (N = 4). (**E**) Percentage live cells following 72 h of incubation in HBSS as analyzed by Trypan Blue exclusion in WT and PC2KO mIMCDs, transfected either with Vehicle (+Veh) or human PC1 (+hPC1). Paired observations of each independent experiment are represented by the same symbol (N = 4). NS: not significant, ^∗^
*p* < 0.05, ^∗∗^
*p* < 0.01.

**Figure 7 ijms-22-13511-f007:**
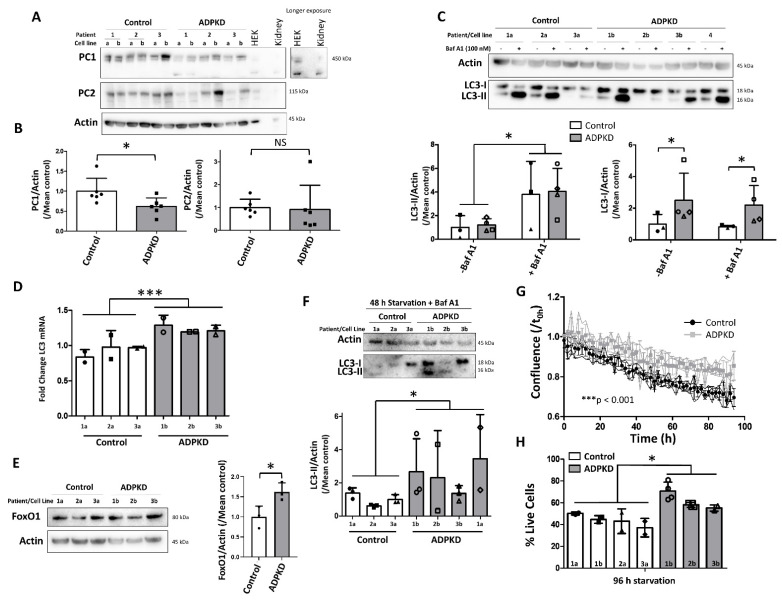
Enhanced starvation-induced autophagy in proximal tubular cell lines derived from ADPKD patients. (**A**) Western blot analysis of the annotated proteins in proximal tubular epithelial cells (PTECs) from 3 young healthy individuals (Control 1–3) and ADPKD patients (ADPKD 1–3). For each individual, 2 monoclonal cell lines were analyzed (a,b). Human embryonic kidney (HEK) cell lysate and lysate of human renal tissue (kidney) were used as controls. (**B**) Quantification of polycystin-1 (PC1) and -2 (PC2) over Actin, normalized to the mean of the control group, in the 6 Control (white) and 6 ADPKD (grey) PTECs. (**C**) LC3 Western blot analysis in 3 PTECs from 3 young individuals (1a, 2a and 3a; white) and 4 cell lines from 4 young ADPKD patients (1b, 2b, 3b and 4; grey). Cells were treated with DMSO (−Baf A1) or 100 nM Baf A1 (+Baf A1) for 3 h before harvest. Upper: representative Western blot. Lower left: quantification of the average LC3-II levels over Actin, normalized to the mean of the control group. Lower right: quantification of the average LC3-I levels over Actin, normalized to the mean of the control group. (**D**) LC3 mRNA expression was evaluated by quantitative PCR (qPCR) in 3 PTECs from 3 young healthy individuals (Control 1a, 2a, 3a; white + closed symbols) and 3 young ADPKD patients (1b, 2b, 3b; grey + open symbols). The symbols represent the 2 independent experiments in each cell line. (**E**) FoxO1 protein levels in 3 PTECs from 3 young healthy individuals (Control 1a, 2a and 3a) and 3 PTECs from 3 young ADPKD patients (1b, 2b and 3b). Left: representative Western blot; right: quantification of FoxO1 levels over Actin (N = 2). (**F**) LC3 Western blot analysis in 3 PTECs from 3 young healthy individuals (1a, 2a and 3a; white + closed symbols) and 4 PTECs from 3 young ADPKD patients (1a, 1b, 2b and 3b; grey + open symbols), subjected to 48 h of starvation. Cells were treated with 100 nM Baf A1 (+Baf A1) for 3 h before harvest. Upper: representative Western blot. Lower: quantification of the average LC3-II levels over Actin of 3 independent experiments in 7 cell lines. Each symbol represents an independent experiment. (**G**) Analysis of confluence of 4 control (black) and 4 ADPKD (grey) PTECs during incubation in HBSS using the IncuCyte Live Cell Analyzer. The curve of each cell line is plotted with the average of each group represented by the superimposed symbols. Linear regression analysis revealed significantly different slopes between Control and ADPKD group. (**H**) Percentage live cells following 96 h of incubation in HBSS as analyzed by Trypan blue exclusion in 4 PTECs from 3 young healthy individuals (1a, 1b, 2a and 3a; white + closed symbols) and 3 PTECs from 3 young ADPKD patients (1b, 2b and 3b; grey + open symbols). Each symbol represents an independent experiment. Throughout this figure, the same individuals/patients are assigned with the numbers 1 to 4, while monoclonal cell lines with letters a and b. NS: not significant, ^∗^
*p* < 0.05, ^∗∗∗^
*p* < 0.001.

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
