# Peer review of "Interdependent Regulation of Polycystin Expression Influences Starvation-Induced Autophagy and Cell Death"

_ijms, 2021, doi:10.3390/ijms222413511_

Round 1

Reviewer 1 Report

It is generaly well written paper presenting laboratory results of experiments on authophagy in ADPKD on a different cells models. Unfortunatelly, there is no translation of these results to the managment of patiens.

I have only few comments.

Introduction

Polyuria is the problem for the patients, and aquaresis the mechanism. It is better to write 'polyuria in the course of aquaresis' or somesing similar.

The progression of the disease cannot be severe. The disease may progress faster leading to the end stage kidney disease in younger age. Please reformulate.

Study aim

It should be reformulated. It is true that polycystin-deficient renal cells have altered cell survival and death during stress. However, in this condition there is no interdependent regulation of PC1 and PC2 expression as one of the gene product is inactive or missing. Therefore it is hard to translate such hypothesis on the ADPKD model.

Reviewer 2 Report

Decuypere and colleagues investigated autophagy and cell death in mouse inner medullary collecting duct cells from wild-type and polycystin-1 (PC1) or polycystin-2 (PC2) knockout mice and proximal tubular epithelial cells (PTEC) from early-stage ADPKD patients with PKD1 mutations versus healthy individuals. Nutrient starvation was induced by incubating the cells in HBSS.

The manuscript is hard to read, important data of the cell line generation are not provided, and largely different time points investigated. Many additional overexpression and siRNA experiments would be required to support the conclusions of the authors.

Some examples of the obvious problems are listed below:

The nomenclature is confusing. The mutations are in the PKD genes encoding for PC1 and PC2.

In the authors list * and $ are confusing/not properly explained.

The manuscript is uploaded as additional unpublished material.

Figure 1 is cut. Only a part is visible and could be evaluated.

Figures 1 A and B are not described in the text. Figure 1 A loading control is different. Figure 1B RNA data should be described in detail. How was the knockout done? Why there is still 30% detectable RNA.

Statements regarding Figure 1C are not visible.

Figure 1D, no significances indicated.

Figure 1E, different time scaling compared to Figure 1D. No significance is indicated.

The “starvation” protocol is obscure as normally authors use defined media composition e.g., high versus low glucose instead of bathing the cells in salt water, which represents not only starvation but an additional removal of all growth factors.

Figure 1 G: loading controls are different again. Differences in Trypan Blue exclusion could not simply be attributed to caspase 3 cleavage.

Description relating to Figure 1H is completely unclear.

Figures 1J,K loading controls missing. General not adequate measure for mTOR activity.

Figure 2 A: Actin levels are different again. GFP and DAPI correspond in 2D but not in 2F.

Figure S1 is not mentioned?

2.3. The statement is not evident from Figure S2.

Figure S2 B, no illustration for 0 hours is provided, but it is quantified afterwards.

“a possible enhanced Sqstm1 degradation” no evidence is provided.

Figure S3 A: actin is saturated. The quantification is not corresponding to the blot.

In Figure 3, the authors use a 48 hour time point, which shows no differences in Figure 1 D,E.

Figure 4: time points seem to be randomly selected again and are different for the subpanels. Actin levels are again all different except for 4C. The PC2 siRNA seems to have no effect.

“However, following 48 h of starvation, PC2KO mIMCDs displayed enhanced autophagy compared to WT, as analyzed with LC3-II Western blotting (Figure 5B) and GFP-LC3 punctae measurements (Figure 5C).” is not obvious from the Figure.

“This was associated with a slightly higher resistance to cell death in PC2KO mIMCDs, most notably following 72 h of starvation (Figure 5D)” is wishful thinking. The only significant difference is observed after 72 h.

Figure 5 F. Hif-1 upregulation can be confirmed only on the protein level. Given the rapid protein degradation, mRNA levels are not sufficient for the conclusion.

Figure 6: actin levels are different. Again, no explanation for the different time points is provided.

Figure 7 C, D: mRNA and protein levels do not correspond very well. Figure 7E: the quantification is not evident from the shown blot. Figure 7F actin levels are totally different again. Figure 7G: the significance corresponds to which time point?

Round 2

Reviewer 1 Report

The authors improved the paper. There are some critical remarks shown by the second reviever, but in my opinion they are not fundamental.

Author Response

Response to Reviewer 1 Comments

The authors improved the paper. There are some critical remarks shown by the second reviever, but in my opinion they are not fundamental.

We thank the reviewer for the positive appraisal of the manuscript. Although the comments made by reviewer 2 were considered not fundamental to this reviewer, we have still attempted to address them (see the response to reviewer 2).

Reviewer 2 Report

The authors argued extensively but did less to improve the manuscript. Figure 1 is visible in the revised manuscript, which represents an improvement. The choice of the different time points was integrated in the response but should be reflected also in the text. The characterization of the cell lines 462ff is unchanged. Sequencing data regarding the frameshift insertions-deletions, a schematic illustration of the resulting protein truncation and position of the PCR primers of quantitative RT-PCRs should be provided.

Author Response

Response to Reviewer 2 Comments

The authors argued extensively but did less to improve the manuscript. Figure 1 is visible in the revised manuscript, which represents an improvement. The choice of the different time points was integrated in the response but should be reflected also in the text. The characterization of the cell lines 462ff is unchanged. Sequencing data regarding the frameshift insertions-deletions, a schematic illustration of the resulting protein truncation and position of the PCR primers of quantitative RT-PCRs should be provided.

- The choice of the different time points was integrated in the response but should be reflected also in the text.

As requested by the reviewer, we have integrated our choice of time points in the manuscript, line 164-168:“However, cell death differences between WT and PC1KO were most clear following several days of starvation. Because autophagy activation typically precedes cell death initiation, we evaluated cell death following 72 h and autophagy following 48 h of incubation in HBSS throughout the remainder of the manuscript.”

  • The characterization of the cell lines is unchanged. Sequencing data regarding the frameshift insertions-deletions, a schematic illustration of the resulting protein truncation and position of the PCR primers of quantitative RT-PCRs should be provided.

Please note that the PC2KO cells were presented before in a peer-reviewed publication in Sci Signal (2019), reference 41 in our manuscript.

We have now inserted a new supplementary figure (Figure S1), illustrating the targeted region of the PKD1 gene and the validation of the KO. We have added the information in the Materials and Methods section, line 443-451). In the manuscript, this figure is referred to in line 67: “A genomic deletion in exon 2-3 of PKD1 in PC1KO cells was confirmed (Figure S1).”

As the cells have multiple copies of PKD1, that each will have different mutations, the sequencing chromatogram of this region will result in an overlap of the different signals (see Figure S1D), and it will not be possible to determine the specific mutations and the resulting protein truncation from this analysis. However, the gel electrophoresis in Figure S1C shows clearly a deletion in all copies.

The exact sequences of the qPCR primers (Mm00465434_m1 for mPKD1 and Mm00435829_m1  for mPKD2, both from ThermoFischer Scientific) are not provided by the manufacturer, but they span exon 1 and 2 for both genes. Please note this is mostly upstream of the sgRNA target sites (which for both is around exon 2 and 3), explaining why some residual expression can be observed in Fig. 1B and Fig. 5A. We have addressed this in line 69-71: “Some residual expression was still observed, most likely because the qPCR primers used target a region mostly upstream (exon 1-2) of the regions targeted by CRISPR-Cas9 (exon 2-3).”